# Predicted Impact of the Lockdown Measure in Response to Coronavirus Disease 2019 (COVID-19) in Greater Bangkok, Thailand, 2021

**DOI:** 10.3390/ijerph182312816

**Published:** 2021-12-05

**Authors:** Sonvanee Uansri, Titiporn Tuangratananon, Mathudara Phaiyarom, Nattadhanai Rajatanavin, Rapeepong Suphanchaimat, Warisara Jaruwanno

**Affiliations:** 1International Health Policy Programme, Ministry of Public Health, Nonthaburi 11000, Thailand; sonvanee.u@ihpp.thaigov.net (S.U.); titiporn@ihpp.thaigov.net (T.T.); mathudara@ihpp.thaigov.net (M.P.); nattadhanai@ihpp.thaigov.net (N.R.); rapeepong@ihpp.thaigov.net (R.S.); 2Bureau of Health Promotion, Department of Health, Ministry of Public Health, Nonthaburi 11000, Thailand; 3Division of Epidemiology, Department of Disease Control, Ministry of Public Health, Nonthaburi 11000, Thailand

**Keywords:** Coronavirus Disease 2019, lockdown, reproduction number, Thailand

## Abstract

In mid-2021, Thailand faced a fourth wave of Coronavirus Disease 2019 (COVID-19) predominantly fueled by the Delta and Alpha variants. The number of cases and deaths rose exponentially, alongside a sharp increase in hospitalizations and intubated patients. The Thai Government then implemented a lockdown to mitigate the outbreak magnitude and prevent cases from overwhelming the healthcare system. This study aimed to model the severity of the outbreak over the following months by different levels of lockdown effectiveness. Secondary analysis was performed on data primarily obtained from the Ministry of Health; the data were analyzed using both the deterministic compartmental model and the system dynamics model. The model was calibrated against the number of daily cases in Greater Bangkok during June–July 2021. We then assessed the outcomes (daily cases, daily deaths, and intubated patients) according to hypothetical lockdowns of varying effectiveness and duration. The findings revealed that lockdown measures could reduce and delay the peak of COVID-19 cases and deaths. A two-month lockdown with 60% effectiveness in the reduction in reproduction number caused the lowest number of cases, deaths, and intubated patients, with a peak about one-fifth of the size of a no-lockdown peak. The two-month lockdown policy also delayed the peak until after December, while in the context of a one-month lockdown, cases peaked during the end of September to early December (depending on the varying degrees of lockdown effectiveness in the reduction in reproduction number). In other words, the implementation of a lockdown policy did not mean the end of the outbreak, but it helped delay the peak. In this sense, implementing a lockdown helped to buy time for the healthcare system to recover and better prepare for any future outbreaks. We recommend further studies that explore the impact of lockdown measures at a sub-provincial level, and examine the impact of lockdowns on parameters not directly related to the spread of disease, such as quality of life and economic implications for individuals and society.

## 1. Introduction

In 2020, the world recognized Severe Acute Respiratory Syndrome Coronavirus-2 (SARS-CoV-2), a viral pathogen that caused one of the most serious health threats in human history, known as Coronavirus Disease 2019 (COVID-19). The first reported case was found in Wuhan, China, after which global cases rose exponentially [1]. To date (June–July 2021), the World Health Organization (WHO) states that the number of COVID-19 cases worldwide has reached over 186,300,000, with 4,028,896 deaths [2].

In late March 2020, the first wave of COVID-19 hit Thailand. The majority of cases originated from superspreading events (SSEs) in nightclubs and boxing stadiums, with the peak of daily cases numbering about 188 [3]. With the rapid control of the disease by strict measures and nationwide lockdown, Thailand was successful in containing the outbreak. The country was widely praised in the global health arena and was ranked among the top successful countries in the world according to the Global COVID-19 Recovery Index [4].

However, the situation changed in 2021 as Thailand faced several further waves of COVID-19 cases and deaths. As a result, the country encountered its most devastating health and economic crisis for many decades. The second wave began in mid-December 2020 and lasted until late February 2021, with a total of 19,867 confirmed cases. Most of the cases at the beginning of the second wave were Myanmar migrant workers in Samut Sakhon—a vicinity province of Bangkok. Some had a history of illegal border crossing without undertaking the quarantine [5].

The situation worsened in April 2021 as the country was hit by a third wave, caused by the Alpha variant of COVID-19 [6]. The outbreak was believed to have originated from an SSE in a nightclub in Bangkok, and then spread to the local communities, prisons, nursing homes, and eventually all over the country. The Thai government responded to this by banning public gatherings (limited to 20 people), closing fitness centers and parks, and banning on-site dining [7], although they stopped short of an official city lockdown.

The fourth wave began in June 2021, after the advent of the Delta variant of COVID-19 [8]. It is widely accepted that the Delta variant is the most transmissible variant (roughly 2–3 times more contagious than the original strain from China). As of the end of July 2021, the Delta variant has been detected in nearly every country in the world [9].

The local outbreak of the Delta variant was first confirmed at a construction site in Bangkok [10]. Within a few weeks, the Delta variant overtook the Alpha variant as the dominant strain in the country (accounting for 69% of infections in mid-July 2021) [11] and accounting for more than 370,000 cases and 3000 deaths, respectively. Many hospitals were reported as facing a shortage of medical equipment and supplies, and it seemed that the Thai healthcare system was going to be overwhelmed by the exponential rise in the cases [12,13].

As a result, the Thai Government decided to take a bold decision by implementing lockdown measures in Greater Bangkok (Bangkok and five associated provinces). The Thai Government later implemented lockdown measures in other provinces where the cases showed a rising trend. The city lockdown did not have any specific time limit, and the Thai Government announced that the end of the measure depended on the assessment of the situation at that time. The Department of Disease Control was assigned by the Thai Government to predict COVID-19 cases, deaths, or intubated patients after the lockdown measure was enforced. The findings would serve as an important input to allow the Thai Government to plan healthcare resources to adequately meet the projected demand. The objective of this study was to assess the severity of the outbreak over the following months given different levels of lockdown effectiveness and different durations of the lockdown.

## 2. Materials and Methods

### 2.1. Study Design

Secondary data analysis was used. The parameters applied in the model were retrieved from a document review of the internal database of the Department of Disease Control (DDC) and the Department of Medical Services (DMS), parts of the Ministry of Public Health. Some basic parameters, such as incubation period and duration of infection, were retrieved from international literature. Parameters that were not available in the literature (for example, time lag from testing to isolation) were obtained from model calibration or expert opinions. Note that for this study, we focused on the situation in Greater Bangkok only. The forecasting started from 19 July 2021 onwards (the day the Thai Government proclaimed a city lockdown).

### 2.2. Model Framework

We adopted a compartmental susceptible-exposed-infectious-recovered (SEIR) model, in combination with the system dynamics (SD) model, as a base framework [14,15]. The simplified model framework is displayed in Figure 1.

The model divided the interested population into different groups: the susceptible, the exposed, the infectious, and the recovered. The speed of transfer from the susceptible group to the exposed group was determined by the effective reproduction number [13]. The transition from being exposed to being infectious was influenced by the incubation period. We modified the traditional SEIR model by dividing the infectious group into before isolation and after isolation. The reason was that once admitted to a hospital, an infected person would be isolated by the hospital protocol [16]. The length of stay (LOS) in a hospital shaped how quickly a patient recovered. The DMS divided infected people into four subgroups (asymptomatic or mildly symptomatic, symptomatic without pneumonia, pneumonia without intubation, and pneumonia with intubation (needing intensive care)) [16]. Some of the intubated cases would die and we assumed that no deaths would occur without intubation.

### 2.3. Model Assumptions

We relied on the following assumptions. First, we assumed that the size of the true population of Greater Bangkok was about 20% larger than the number reported in the civil registry [17]. This is because Greater Bangkok is the economic capital of the country where people from upcountry and migrants came to work (without changing their residential address). The total volume of Greater Bangkok in the civil registry was about 10 million, but for this study, we extrapolated to reach a number of 12 million.

Second, in practice, during the pandemic, the number of initial infected people could not be measured precisely. We therefore obtained this number from model calibration, as well as the value of the effective reproduction number. During 1–15 June 2021, based on the DDC internal monitoring, the effective reproduction number in Bangkok was equal to 1.1. However, when we used this value to estimate the number of daily cases between 16 June and 18 July 2021, the model produced a lower number than the number of cases actually reported. To address this, a third assumption was applied. Third, we postulated that there was some degree of underreporting of asymptomatic and mildly symptomatic cases. The literature suggested that underreporting was a common phenomenon in many countries, even those with advanced health systems [18]. This was affirmed by the field operation of the Rural Doctor Society of Thailand in July 2020, reporting that over 13% of people in 40 high-density communities in Bangkok were found infected during COVID-19 screening [15]. However, we hypothesized that there was no underreporting of the intubated cases and deaths.

Concerning the second and third assumptions, we calibrated the reproduction number and the percentage of severity distribution against the daily death toll. We found that the estimates based on the reproduction number of 1.43 (about 30% larger than the reported reproduction number) and the percentage of initial infected people per total population of 0.5% fitted the death toll best. Using a reproduction number of 1.43 suggested that the actual numbers of asymptomatic and mildly symptomatic cases were about three to six times greater than the reported numbers. More information on the percentage of clinically severe cases and other relevant parameters are demonstrated in the later subsection, “Essential parameters and formula”.

### 2.4. Model Scenarios and Outcomes of Interest

We attempted to address the policy question as to what the situation would look like 90–120 days after implementing the lockdown policy. The official announcement to the public suggested that the lockdown policy was likely to be in effect for about 30–60 days. During the first wave of COVID-19 in Thailand, in early 2020, Leerapan et al. estimated that the lockdown helped reduce the reproduction number by about 20% [19]. We postulated that the effect of the lockdown measure might vary between 20 and 60% as, this time, there were many more additional measures compared with the first wave, such as mass COVID-19 vaccination campaigns and active community screening. At the time of writing, the COVID-19 vaccine campaign in Bangkok was underway and there was no significant change in the vaccine campaigns between before and after the lockdown. Thus, we assumed that the case fatality ratio would not significantly change in a short period, and for simplicity, we postulated that the lockdown would affect the reproduction number only.

With all accounts above, we compared the following outcomes—(i) daily reported incident cases, (ii) daily actual incident cases, (iii) prevalent intubated cases, and (iv) daily deaths—in different scenarios as follows, Table 1.

### 2.5. Essential Parameters and Formula

We used Microsoft Excel and Stella 2.0 (number: 251-401-786-859) to run the model. Table 2 and Table 3 demonstrate some important parameters and formulas applied in the model.

### 2.6. Ethics Consideration

As this study relied only on publicly available secondary data and did not involve human participation, ethics approval was not needed. Nonetheless, we strictly adhered to ethical research standards by not presenting the results in a way that could be traced back to individual information.

## 3. Results

We first presented the number of estimated actual daily incident cases in Figure 2. The pattern of the daily incident cases was similar in all scenarios (except the no-lockdown scenario). The no-lockdown policy demonstrated the largest daily case toll compared with other scenarios. The peak of daily cases reached over 100,000 per day from late August to early September and dropped to lower than 20,000 per day by late October. Policy scenarios with a one-month lockdown caused a suppression of the incident cases until mid-August, and then the case volume rebounded, peaking at 70,000–80,000. The scenario with 20% lockdown-effectiveness reached its peak about a month earlier than the 40% lockdown-effectiveness and two-months earlier than the 60% lockdown-effectiveness scenario. For the two-month lockdown, the daily case toll was suppressed to below 20,000 a day until late October for the 40%- and 60%-effectiveness scenarios. The two-month lockdown policy with 60% effectiveness showed the lowest daily case toll relative to all other policy scenarios (lower than 10,000 cases a day until early December).

The daily reported cases in Figure 3 followed the same pattern as the actual daily (estimated) case toll, but the magnitude was about three quarters smaller. The daily reported cases in the no-lockdown policy exceeded 25,000 during mid-September and then steadily declined to below 5000 by late October. For the two-month lockdown policy with 20% effectiveness in reproduction number reduction, the case toll plateaued at about 7000–8000 during early August to mid-September, slightly increased to 1100 in late October, and then declined to just above 5000 in early December. The one-month policy scenarios displayed the peak of cases by early October (~18,500 in the 20%-effectiveness scenario), early November (~16,000 in the 40%-effectiveness scenario), and early December (~16,000 in the 60%-effectiveness scenario).

Daily deaths numbered over 300 from mid-September to early October in the no-lockdown policy scenario. The second-highest toll (almost 250 deaths) was found in a one-month lockdown policy with 20% effectiveness, though the peak was delayed about a month compared to the no-lockdown scenario. The one-month scenarios with 40% and 60% effectiveness displayed almost the same peak magnitude (~200–250 deaths) but with peaks about a month apart (mid-October for the 40%-effectiveness scenario and mid-November for the 60%-effectiveness scenario). All of the two-month lockdown scenarios demonstrated a death toll below 150 till mid-November, Figure 4.

The prevalence of intubated patients followed a similar pattern to that of deaths. The two-month lockdown scenario with 60% effectiveness resulted in the lowest number of intubated cases, with about 40 cases a day during late October. Apart from the no-lockdown policy, all policy scenarios showed a peak of intubated cases that stayed below 2000. The peak timing for respiratory ventilator demand (as reflected by the prevalence of intubated patients) varied across scenarios, such as mid-November for the one-month lockdown with the 40% effectiveness scenario (peak magnitude~1750) and the two-month lockdown with the 20% effectiveness scenario (peak magnitude~1300), Figure 5.

## 4. Discussion

Overall, this study provides a clear insight into the range of possible COVID-19 outbreak situations in Thailand, given varying durations and effectiveness of the lockdown policies. A lockdown can delay the peak of new cases and deaths and help “buy time” for the health system to prepare resources, including beds and ventilators, ensuring that case numbers do not exceed health system capacity [22]. Policymakers may need to consider implementing “harsh measures” in terms of reproduction number reduction.

We also found that after the lockdown was released, it was very likely that the numbers of cases and deaths would bounce back. This finding is in line with the theory that the epidemic will not die out as long as the percentage of the remaining susceptible population is larger than the inverse of the reproduction number [23]. In other words, a scenario with a large reproduction number (no lockdown) is likely to demonstrate a quick rise and fall in the cases and deaths, while a lockdown scenario (which reduces the reproduction number) may delay the peak and risks a new rise in cases once the lockdown measure is relaxed. Therefore, policymakers face a dilemma: whether to implement a “harsh lockdown” in a short period or a “soft lockdown” for a long period. Our findings suggest that the two-month lockdown scenario with 60% effectiveness would be able to control the outbreak for over five months. However, the literature has suggested that, during the first wave of the COVID-19 epidemic in Thailand, lockdown policy was able to reduce the reproduction number by 20–25% at most. Hence, we believe a 60% effective scenario is unlikely to be obtainable, and a more realistic approach is to use a lockdown policy as a measure to slow the peak, and to prevent the health system from collapsing. A large wave of cases leads inevitably to hospitalization in severe cases and death [24], and traditional health service systems were not designed to respond to the abrupt demands of a pandemic. Hospital bed shortages, especially intensive care beds that are normally needed for intubated patients, require both large medical resources and health personnel. In April 2021, total bed availability in Greater Bangkok was about 20,000 with 1050 ICU beds. By September 2021, this expanded to 54,000 beds, including 1650 ICU beds [25]. Home isolation and community isolation measures should be considered to minimize the burden on hospitals and to prevent hospitals from overcrowding as, before the lockdown was implemented, all infected cases were admitted to a health facility, following the treatment guidelines of the Department of Medical Services [26]. The Thai Government should implement a guideline to specify who should be treated in nonfacility settings. Many countries such as the United Kingdom, United States of America, or Germany have implemented the home isolation system for mild cases while hospital beds are kept for more severe cases.

The timing of lockdown initiation is also critical. The later the start, the higher the caseload and the longer the lockdown required to prevent the health system from being overwhelmed [24]. Normally, outbreak control relies on individual contact tracing to minimize and avoid outbreaks. However, when cases far outnumber the capacity of contact tracing and investigation, the objective is instead to minimize the damage to the national public health as a whole.

However, a lockdown is not a silver bullet as it will always sacrifice the national economy. In the first wave of the epidemic, Thailand decided to implement a nationwide lockdown in March 2020, while cases were below 200 a day. As a result, a heavy economic downturn was evident (−6% of the gross domestic product (GDP) in 2020). This might be one of the key reasons why the Thai Government implemented the nationwide lockdown quite late in the fourth wave. Thus, the bottom line is not just implementing or not implementing the lockdown, but it is, if the lockdown is to be implemented, how the Thai Government maintains the national economy (or at least minimizes the economic damage). A recent study from Israel suggests that a cyclical lockdown strategy (four days of complete lockdown alternating with ten days of reopening) was effective at halting the epidemic while limiting the negative effects on the economy [19]. However, such a proposal might not be practical in reality, as it significantly disrupted the working life of most people. Financial relief or other monetary support should be implemented alongside the lockdown as this can help the economy recover [27], as well as encourage people to comply with the lockdown, especially the self-employed and vulnerable populations [22]. The World Bank suggested that employment retention policies, subsidies for employees, wage support for affected businesses, co-payment of new hires (specifically new graduates) to prevent long-term unemployment, and training support for nonformal sectors can help ensure medium-term and long-term economic sustainability [28].

The acceleration of vaccine rollout during the lockdown period is essential to contain the outbreak and reduce the death toll. The literature suggests that all vaccine types can avert new infection, death, and peak hospitalization when the reproductive number is low (below 1.8) with at least 50% coverage and a rapid pace of vaccine rollout [29]. Since the advent of the fourth wave in mid-2021, the vaccine rollout in Thailand had been sped up significantly. It took four months to vaccinate 10 million doses (March–June), compared with 21 million doses during (July–August).

Other public health measures are also indispensable. These include finding, testing, tracing, isolating, supporting (FTTIS), alongside the lockdown policy. Otherwise, the case toll may likely escalate again once the lockdown is released, as suggested by the findings above. FTTIS measures and the rapid expansion of vaccine coverage will reduce the susceptible population and reduce the contact frequency and contact probability between infected and susceptible people. This should reduce cases and deaths in the long run [30]. A study in France found that most COVID-19 cases were undetected, especially when the incidence was high. Of the ten symptomatic cases, nine were missed in the surveillance system [31]. Just before the lockdown in July, Reverse Transcriptase Polymerase Chain Reaction (RT-PCR)-positive rates in Thailand had reached 10%, almost double the positive rate (~5%) at the beginning of the third wave in April, implying that the testing burden sharply increased [32]. The Thai Government therefore proposed the use of the Antigen Detection Rapid Diagnostic Test (Ag-RDT). The Ag-RDT can play an important role, particularly when the cases are increasing rapidly. At the time of writing, the Thai Government bought millions of Ag-RDTs to facilitate access to testing and reducing the workload of staff at RT-PCR testing facilities. However, the challenge lies in the issue of whether these tests will be adequately (and equitably) distributed to persons in need. This demands continuous monitoring and evaluation of the distribution of Ag-RDTs [33].

Self-protecting behavior is always key for long-term suppression of the reproduction number, finally resulting in a flattening of the epidemic curve containment. These include mask-wearing, physical distancing, and handwashing with alcohol-based sanitizers [31,34]. However, these nonpharmacological interventions (NPIs) always come with behavioral fatigue, and not all people can comply with NPIs all the time, especially for physical distancing [35].

There remain some limitations in this study. First, during the lockdown period, many submeasures were implemented simultaneously, for instance, prohibition of mass gatherings, closing public spaces, work-from-home policy for many businesses, and mandatory face masks in public areas. Thus, it is difficult to distinguish the effect of lockdown (in the form of reproduction number reduction) from that of the different submeasures. Moreover, the term lockdown is used differently in different settings. In this study, we postulated that the lockdown measure tackled the reproduction number directly. The readers should be aware that in some countries, the term lockdown may come together with a rapid detection of cases. Translating this into the SEIR framework, the lockdown measure may alter the time lag between before isolation and being isolated. In an extreme scenario, if the lockdown means a rapid detection of cases, all cases are detected soon enough to prevent the transformation of mild to severe cases. Such a scenario means the alteration of the clinical profile parameters, which ultimately determines the rate of propagation from infectious compartment to recovery compartment. Hence, the readers should be aware of the nuanced difference in the term lockdown across the literature. Second, the model utilized macro-level data, and the nature of the analysis always assumes homogenous contact within a unit of interest. In other words, we postulated the same severity of the epidemic throughout Greater Bangkok for each policy scenario. However, in reality, if we explore the results granularly, there are always varying degrees of the epidemic across districts or subdistricts. Third, the assumption that no deaths occur outside the hospitals may not be true, especially in the countries where the policy allows the patients to be self-treated. To be more specific, the probability of death is not solely due to the natural course of disease itself but depends on the health system on which each country relies. However, in this setting, the Thai Government requires all COVID-19 cases to be hospitalized (regardless of the severity status). The internal monitoring of the DDC informed that during the peak period of the cases, the percentage of COVID-19 deaths outside health facilities was 3.9% (compared with all deaths), and this figure became much smaller outside the peak period. This implies that the death toll of this study might be slightly underestimated up to 4% during the peak when the healthcare capacity was extremely stretched. Lastly, the model depends on numerous assumptions, such as a fixed reproduction number, and constant policy effectiveness. In reality, there are many more unobserved (or uncontrolled) factors that can influence the epidemic, such as viral mutation, unexpected superspreading events, and social and political pressure. Further studies that explore the impact of lockdown measures at sub-provincial levels and that examine the impact of lockdown on other aspects of life, including the economic consequences for individuals and society, would be of great value.

## 5. Conclusions

The duration of lockdown and the effectiveness of the lockdown policy in diminishing the reproduction number resulted in different peaks of daily new cases, daily deaths, and intubated patients. The implementation of a lockdown policy did not eliminate the epidemic but rather delayed the peak of cases to enable the health system to mobilize more resources to address demand. During a lockdown, other supportive measures should be in place. These include the acceleration of FTTIS, and of vaccine rollout, and adequate and equitable distribution of Ag-RDT. Additional research that delves into the impact of lockdown measures in more granularity and investigates the impact of lockdown on the people’s quality of life and economic consequences for society more widely, would also be useful.

## Figures and Tables

**Figure 1 ijerph-18-12816-f001:**
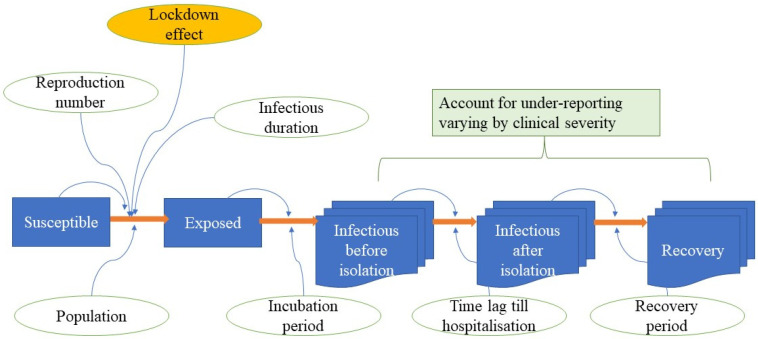
Model framework.

**Figure 2 ijerph-18-12816-f002:**
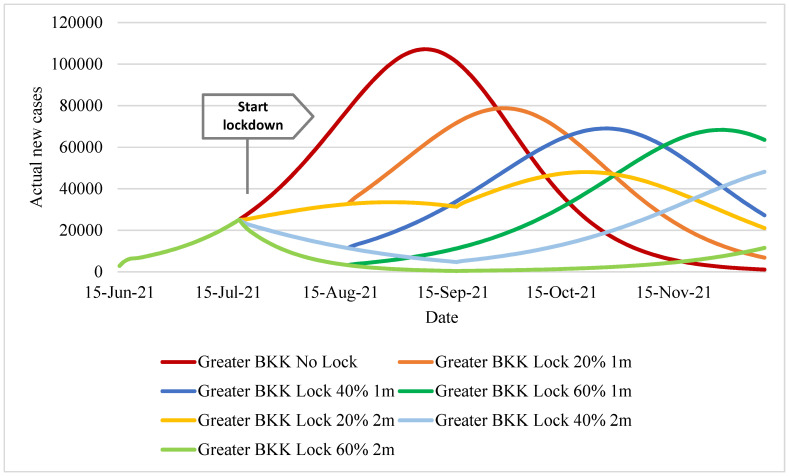
Daily (estimated) actual new cases in Greater Bangkok.

**Figure 3 ijerph-18-12816-f003:**
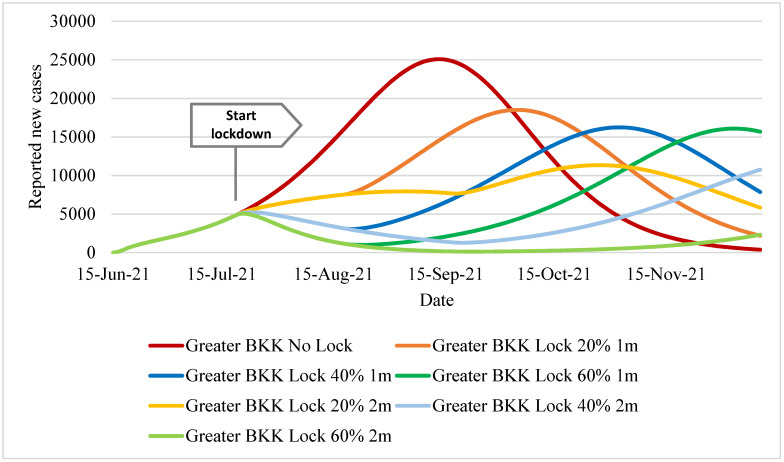
Daily new reported cases in Greater Bangkok.

**Figure 4 ijerph-18-12816-f004:**
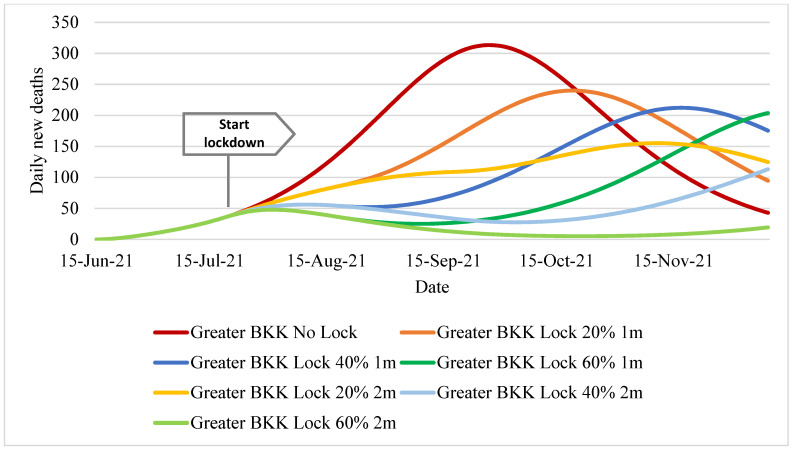
Daily new deaths in Greater Bangkok.

**Figure 5 ijerph-18-12816-f005:**
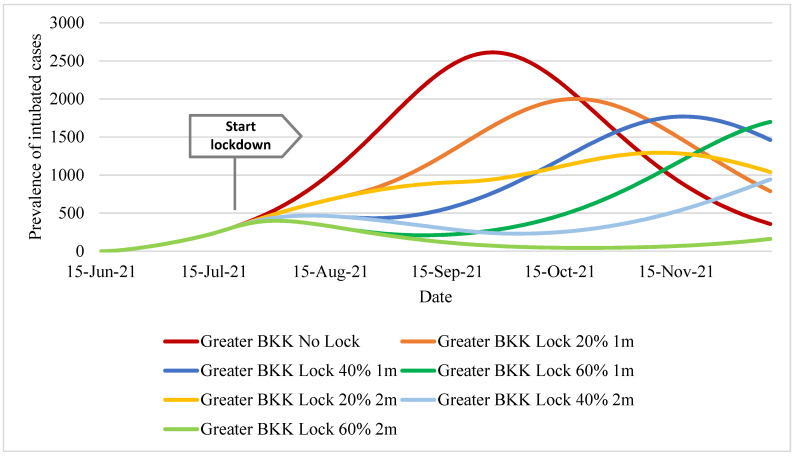
Prevalent reported intubated cases in Greater Bangkok.

**Table 1 ijerph-18-12816-t001:** Scenarios of interest.

Scenario	Lockdown Effect (% Reduction in Reproduction Number)	Duration of Lockdown (Months)
1	No lockdown (theoretical reference)	
2	20	1
3	40	1
4	60	1
5	20	2
6	40	2
7	60	2

**Table 2 ijerph-18-12816-t002:** List of key parameters.

Parameters	Unit	Value	Reference
Reproduction number	Dimensionless	1.43	Model calibration
Population in Greater Bangkok	Persons	12,200,000	National Statistical Office of Thailand [20] and model estimation
Percentage of initial infectees per total population	Dimensionless	0.5	Model calibration
Average infectious duration	Days	5	Ganyani et al. [21]
Average incubation period	Days	5.2	McAloon et al. [22]
The time lag from being infected to isolation	Days	5	Model calibration
Percentage of reported asymptomatic and mildly asymptomatic cases	Dimensionless	42.6	Internal database of the Department of Disease Control
Percentage of reported symptomatic nonpneumonic cases	Dimensionless	54.2	Internal database of the Department of Disease Control
Percentage of reported symptomatic pneumonic cases without intubation	Dimensionless	2.6	Internal database of the Department of Disease Control
Percentage of reported symptomatic pneumonic cases with intubation	Dimensionless	0.6	Internal database of the Department of Disease Control
Percentage of actual asymptomatic and mildly asymptomatic cases	Dimensionless	60.7	Internal database of the Department of Disease Control
Percentage of actual symptomatic nonpneumonic cases	Dimensionless	38.6	Internal database of the Department of Disease Control
Percentage of actual symptomatic pneumonic cases without intubation	Dimensionless	0.6	Internal database of the Department of Disease Control
Percentage of actual symptomatic pneumonic cases with intubation	Dimensionless	0.1	Internal database of the Department of Disease Control
Length of hospital stay for asymptomatic and mildly symptomatic cases	Day	14	Internal database of the Department of Disease Control
Length of hospital stay for symptomatic nonpneumonic cases	Day	14	Internal database of the Department of Disease Control
Length of hospital stay for pneumonia cases with and without intubation	Day	21	Internal database of the Department of Disease Control
The ratio of daily incident deaths per prevalent intubated cases	Dimensionless	0.12	Internal database of the Department of Medical Services

**Table 3 ijerph-18-12816-t003:** The essential formula of the model.

Change of Status	Formula	Note
From susceptible to exposed	−β × (1 − κ) × S × I_2_/P	β = reproduction number/infectious duration, κ = lockdown effect, S = susceptible population, I_2_ = isolated infectious population, P = total population
From susceptible to nonisolated infectious	−αE	α = 1/incubation period, E = exposed population
From nonisolated infectious to isolated infectious	−δI_1_	δ = 1/time lag from nonisolation to isolation, I_1_ = nonisolated infectious population
From isolated infectious to recovered	−ζI_2_	ζ = 1/length of stay (varying by clinical severity); I_2_ = isolated infectious population

## Data Availability

The datasets generated and/or analyzed during the current study are not publicly available due to the Thai-DDC’s regulation but are available from the corresponding author on reasonable request.

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
