# Peer review of "Predicted Impact of the Lockdown Measure in Response to Coronavirus Disease 2019 (COVID-19) in Greater Bangkok, Thailand, 2021"

_ijerph, 2021, doi:10.3390/ijerph182312816_

Round 1

Reviewer 1 Report

The governments of many countries established blockades to prevent Covid-19 from spreading.

There are different types of locks.

The authors must present a theoretical framework with different blocking possibilities, compare with those established in their country and discuss the possible results if other blocking measures had been taken.

Author Response

Thank you for your comment. We agree that the term lockdown varies across settings. A brief discussion on this point was added in line 324-333.

Reviewer 2 Report

The authors present forecasting data on the impact of lockdown for the fourth wave of the COVID-19 pandemic in Bangkog, Thailand, based on a SEIR model. They concluded that implementing a short "hard" lockdown would not prevent the surge but delay the peak, which would be likely more effective that a soft prolonged lockdown. 

Modelling studies on COVID-19 and epidemics in general that add to the current knowledge to refine the methods and the granularity of data worlwide are welcome. The present study is interesting and scientifically sound. Authors acknowledge the main limitations of their study. 

I only have one major comment: I do not agree with the assumption that no deaths might occur without intubation. Patient could be not considered candidates for intubation due to several reasons. Moreover, there is a small proportion of patients that might die due to covid-19 but not because of respiratory insufficiency. 

Another implicit assumption is that all patients would die during hospitalization, which is also not true. Could you please confirm this?

Author Response

Thank you for pointing this out. We added a couple of sentences to remind the readers about the possible underestimation of the results if there were cases dying outside health facilities. However, the internal report of the DDC showed that less than 4% of the deaths occurring outside health facilities. We added this point in line 338-348.

Reviewer 3 Report

The article presents a correct condition in terms of its writing and content. Offers highly polished document condition.

Author Response

Thank you so much.